# The Relationship between Obesity-Related Factors and Graves’ Orbitopathy: A Pilot Study

**DOI:** 10.3390/medicina58121748

**Published:** 2022-11-29

**Authors:** Ching Lu, Chao-Lun Lai, Chih-Man Yang, Karen Chia-Wen Liao, Chie-Shung Kao, Tien-Chu Chang, Ming-Der Perng

**Affiliations:** 1Institute of Molecular Medicine, National Tsing Hua University, Hsin-Chu 300044, Taiwan; 2Department of Internal Medicine, National Taiwan University Hospital Hsin-Chu Branch, Hsin-Chu 300195, Taiwan; 3Department of Laboratory Medicine, National Taiwan University Hospital Hsin-Chu Branch, Hsin-Chu 300195, Taiwan; 4Institute of Molecular Medicine and Bioengineering, National Chiao Tung University, Hsin-Chu 300193, Taiwan; 5Biological Sciences Division, University of Chicago, Chicago, IL 60637, USA; 6Department of Ophthalmology, National Taiwan University Hospital Hsin-Chu Branch, Hsin-Chu 300195, Taiwan; 7Department of Internal Medicine, College of Medicine, National Taiwan University Hospital, Taipei 100225, Taiwan

**Keywords:** Graves’ orbitopathy, body mass index, fasting plasma insulin, homeostasis model assessment-estimated insulin resistance

## Abstract

*Background and Objectives:* The aim of this study was to investigate the relationships between obesity-related factors including body mass index (BMI), diabetes or prediabetes, hyperlipidemia, fasting plasma glucose, fasting plasma insulin, homeostasis model assessment-estimated insulin resistance (HOMA-IR), highly sensitive C-reactive protein (hs-CRP) and Graves’ orbitopathy (GO). *Materials and Methods:* Eighty-four patients with Graves’ disease (GD) (42 without GO and 42 with GO) were enrolled in this cross-sectional cohort study. Gender, age, GD treatment history, height, body weight, waist circumference, smoking status, co-morbidities, levels of free thyroxin, thyroid-stimulating hormone, thyroid-stimulating hormone receptor (TSHR) antibodies, fasting plasma glucose and insulin, and hs-CRP were recorded. The eye condition was evaluated using the consensus statement of the European Group of Graves’ Orbitopathy (EUGOGO) and the NOSPECS classification. *Results:* In this study, multivariate regression analysis showed that BMI, fasting plasma insulin, and HOMA-IR were associated with the presence of GO after adjusting the age, gender, smoking, TSHR antibodies, and steroid usage (adjusted odd’s ratio (aOR) 1.182, 95% confidence interval (95% CI), 1.003–1.393, *p* = 0.046; aOR 1.165, 95% CI, 1.001–1.355, *p* = 0.048; and aOR 1.985, 95% CI, 1.046–3.764, *p* = 0.036, respectively). In addition, BMI, fasting plasma glucose, fasting plasma insulin, HOMA-IR, and hs-CRP levels were positively correlated with the severity of GO. *Conclusions:* The findings of this study suggest that obesity-related factors, especially fasting plasma insulin and HOMA-IR, are related to GO. Our study highlighted the importance of obesity-related factors in GO. Obesity-related factors may cause the development of GO or occur simultaneously with GO.

## 1. Introduction

Graves’ orbitopathy (GO), also known as thyroid-associated orbitopathy, thyroid eye disease, and Graves’ ophthalmopathy, is common in patients with Graves’ disease (GD), with an incidence rate of approximately 25–50% for Graves’ disease patients and 5–6% having severe disease. GO is diagnosed in 70% of patients with Graves’ hyperthyroidism by computed tomography or magnetic resonance imaging [1,2,3,4]. The risk factors for GO include smoking, thyroid-stimulating hormone receptor (TSHR) antibody increase, radioactive iodine therapy, high cholesterol level, and thyroid dysfunction [1,2,3,4,5,6,7,8]. The pathogenesis of GO involves hereditary factors, environmental factors, thyroid-stimulating hormone receptor (TSHR) antibodies, and type 1 insulin-like growth factor receptors (IGF 1-R) [1,2,3,5,6,7,8,9,10]. Obesity is a risk factor for and related to the progression of several autoimmune diseases [11], and it is known to induce type 2 diabetes, insulin resistance, and dyslipidemia [12]. A high body mass index (BMI) has been associated with highly sensitive C-reactive protein (hs-CRP) [13], and insulin resistance has been associated with thyroid dysfunction [14] and is higher in patients with vitiligo and rheumatoid arthritis [15,16,17]. Diabetes has also been reported to be a risk factor for GO [18,19,20], which along with obesity and insulin resistance are states of increased inflammation [1,21]. Furthermore, a previous study reported an association between obesity and GO in diabetic patients [20]. However, the relationship between obesity-related factors and GO has yet to be elucidated in GD patients with and without diabetes mellitus. Therefore, the aim of this study was to investigate the relationship between obesity-related factors and GO for understanding the pathophysiology and exploring possible prevention and treatment for GO.

## 2. Materials and Methods

### 2.1. Study Design and Setting

This was a cross-sectional study conducted at one regional teaching hospital in northern Taiwan. 

### 2.2. Subjects

We enrolled 84 patients who visited our internal medicine and ophthalmology department and were diagnosed as having GD between October 2012 and December 2014. Graves’ disease was diagnosed based on clinical symptoms and signs with high TSHR antibodies [1]. The inclusion criteria were as follows: aged between 20 to 80 years, received antithyroid drug treatment for at least 6 months. The exclusion criteria were as follows: with inflammatory eye disease, high myopia, orbital tumor, myasthenia gravis or other eye condition mimicking GO. The study followed the tenets of the 1964 Declaration of Helsinki and its later amendments or comparable ethical standards, and the study protocol was approved by the Institutional Review Board of National Taiwan University Hospital, Hsin-Chu Branch with approval code 100-027-F on 28 September 2012. Written informed consent was obtained from all participants after explaining the nature and design of the study. 

### 2.3. Measurements

Demographic data including gender, age, height, body weight, waist circumference, history and duration of treatment for GD, smoking status, eye surgery, and other systemic diseases were recorded. Levels of free thyroxin and thyroid-stimulating hormone were measured using a chemiluminescent microparticle immunoassay analyzer (ARCHITECTi2000, Abbott, Chicago, IL, USA). TSHR antibodies were measured using a radioimmunoassay with a WIZARD Automatic Gamma Counter (WALLAC 2470, Perkin Elmer, Waltham, MA, USA), and hs-CRP was measured using an immunoturbidimetric assay (7180E, Hitachi, Tokyo, Japan). Fasting plasma glucose was measured using a hexokinase assay (7180E, Hitachi, Tokyo, Japan), and insulin was measured using an enzyme-linked immunosorbent assay (Cortez, Los Angeles, CA, USA). Based on the cross-sectional study design, all data were collected once. All blood samples were drawn from the patients’ arms after fasting for at least 8 h. Some of the TSHR antibody results were obtained from previous medical records; however, all data were recorded within 3 months of the eye assessment. Insulin resistance was calculated using the homeostasis model assessment-estimated insulin resistance (HOMA-IR) equation [22]. BMI was calculated as body weight in kilograms divided by the square of height in meters [23]. Patients were considered to have diabetes mellitus if they had a history of diabetes mellitus and were receiving medicine for the disease. Those with a history of pre-diabetes or a fasting plasma glucose level between 100–125 mg/dl were considered to have pre-diabetes [24]. Central obesity was defined as a waist circumference ≥90 cm in males and ≥80 cm in females [25]. If the patients had high or low levels of free thyroxin, they were defined as having thyroid dysfunction [8]. The definition of smoking included every smoking and passive smoking [8]. Patients were considered to have hypertension and hyperlipidemia if they had a history of the diseases.

Graves’ Orbitopathy (GO) was diagnosed based on the criteria of the European Group on Graves’ Orbitopathy (EUGOGO) Consensus Statement [5]. We also classified the patients by severity according to the EUGOGO classification of severity in GO [5] and the NOSPECS classification [26]. The degree of proptosis was evaluated using a Hertel exophthalmometer and was defined as exophthalmos ≥ 18.6 mm or inequality of proptosis more than 2 mm in the Chinese population [27]. All our patients did not have sight-threatening GO. Some patients whose eye defects had been corrected by surgery for GO before undergoing our evaluations were regarded as having moderate to severe GO [5]. The methods of surgery included orbital decompression, lid retraction correction, and eye muscle surgery. According to the statement of EUGOGO, if the patients have lid retraction > 2 mm, exophthalmos > 21.6 mm, moderate to severe soft tissue involvement, and inconstant or constant diplopia, we defined them as having moderate to severe GO [5]; otherwise, they were defined as having mild GO [5]. See Appendix A for details.

### 2.4. Statistical Analysis

Demographic, clinical, and laboratory data were compared using the independent *t*-test or the Mann–Whitney U test for continuous variables; and we used the chi-square test or Fisher’s exact test for categorical variables. We also calculated Spearman’s correlation coefficients to evaluate the correlations between obesity-related factors and the severity of GO. Multivariate binary logistic regression analysis was used to investigate the associations between obesity-related factors and the presence of GO with adjustments for gender, age, smoking, TSHR antibodies, and steroid use. We regarded a *p*-value < 0.05 to be statistically significant. All statistical analyses were performed using IBM SPSS statistical software for Windows (Version 20.0; Armonk, NY, USA: IBM Corp). (Authorization date 4 July 2012) (Verify license code SP 101070402A).

## 3. Results

Eighty-four patients with GD (42 without GO and 42 with GO) were included in this study. Demographic data, clinical characteristics, and laboratory data were compared between patients with and without GO (Table 1). All cases received antithyroid drug treatment for Graves’ disease for at least half years (6.1 ± 6.7 years) (Table 1). The most obese patient had BMI 31, but most cases were not obese (22.7 ± 3.2) (Table 1). Patients with GO had a higher BMI, levels of fasting plasma glucose, HOMA-IR, and hs-CRP, and they were more likely to have diabetes or pre-diabetes and hyperlipidemia compared to those without GO (Table 1). In the multivariate logistic regression model, BMI (adjusted odds ratio (aOR) 1.182, 95% confidence interval (CI) 1.003–1.393, *p* = 0.046), fasting plasma insulin (aOR 1.165, 95% CI 1.001–1.355, *p* = 0.048), and HOMA-IR (aOR 1.985, 95% CI 1.046–3.764, *p* = 0.036) were associated with the presence of GO, after adjustments for age, gender, smoking, the titer of TSHR antibodies and steroid use (Table 2). Moreover, BMI, fasting plasma glucose, fasting plasma insulin, HOMA-IR, and hs-CRP levels were positively correlated with the severity of GO (rho = 0.285, *p* = 0.009; rho = 0.298, *p* = 0.006; rho = 0.243, *p* = 0.026; rho = 0.270, *p* = 0.013; rho = 0.299, *p* = 0.006, respectively) (Table 3).

There were no cases of Myasthenia gravis, orbital myositis, orbital tumor, high myopia, or other conditions that mimic Graves’ orbitopathy. Patients also had no inflammation disease, which will increase hs-CRP and HOMA-IR in the study.

## 4. Discussion

Most Graves’ ophthalmopathy developed 6 months before or after the diagnosis of Graves’ disease [28], so we enrolled our patients with Graves’ disease in of 6 months of treatment. There are some ocular conditions including idiopathic orbital inflammation, pseudotumor, orbital lymphoma, obesity, Cushing’s syndrome, and myasthenia gravis that mimic Graves’ ophthalmopathy [29,30]. To reduce these biases, we did not include these patients except obesity. Prior studies have suggested the risk factors for GO include smoking, radioactive iodine therapy, high TSHR antibodies, cholesterol level, and thyroid dysfunction. [1,2,3,4,5,6,7,8,9,10,31,32,33,34,35,36,37] In this study, we did not find an association between smoking and GO in Table 1. This was probably the effect of smoking that was confounded by other variables, such as age and gender. After adjusting the age group and gender, smoking became a significant risk factor (not shown). In addition, radioactive iodine therapy and thyroid dysfunction were not associated with GO in the present study. This may be due to the small number of patients who received radioactive iodine therapy and had abnormal thyroid function. Another possibility is that the patients with GD had a relatively stable thyroid function after treatment for at least 6 months. Moreover, in the multivariate regression analysis, the titer of TSHR antibodies was also an independent factor associated with GO in this study (aOR: 1.020; 95% CI 1.000–1.039, *p* = 0.047) (See Appendix A for a comprehensive analysis).

The relationship between obesity and exophthalmos or GO had been investigated in a few studies but not fully elucidated [20]. A previous study demonstrated that higher BMI was associated with a higher degree of proptosis, and enlarged medial rectus muscle diameter in patients without GD. In addition, bilateral exophthalmos was noted in 30% of the obese patients (defined as a BMI ≥ 30 kg/m^2^) and some of the obese patients had eye signs resembling those of patients with GO [38]. Another study reported that a patient with rapid body weight gain without other reasons (e.g., GO) had exophthalmos with global luxation [39]. Moreover, the severity of GO has been found to correlate with a BMI ≥ 26 kg/m^2^ in a recent study among diabetic patients [20]. In agreement with this, we found that BMI was associated with the presence of GO (*p* = 0.034) and positively correlated with the severity of GO (*p* = 0.009). The association between obesity and exophthalmos may be mediated through steroid-induced obesity [40]. However, in this study, BMI remained associated with the presence of GO after adjustments for age, gender, smoking habits, titers of TSHR antibodies, and steroid use (aOR 1.182, 95% CI 1.003–1.393, *p* = 0.046). Moreover, we found a correlation between BMI and the severity of GO. These are compatible with previous studies on patients with GO [20].

Patients with GO were more likely to have hyperlipidemia in this study. This finding was consistent with previous reports that found the association between high total cholesterol and low-density lipoprotein (LDL) cholesterol and the presence of GO [41,42]. Dyslipidemia was also the clinical manifestation of obesity and insulin resistance [43].

Previous studies reported that diabetes was associated with the development and outcomes of GO. One retrospective 5-year study of GO demonstrated that patients with concomitant diabetes mellitus had a higher chance of having a worse visual prognosis compared to those without [18]. One Korean study also found that patients with diabetes mellitus had a more severe course of GO [19]. One retrospective study on the relationship between diabetes and GD or GO demonstrated that (1) GO was associated with type 2 diabetes mellitus, but not type 1 diabetes mellitus; (2) the severe form of GO occurred more frequently in patients with type 2 diabetes mellitus than those without; (3) type 2 diabetes mellitus was the most important risk factor for the severe form of GO; and (4) the duration of diabetes, obesity, and vascular complications were associated with the severe form of GO [20]. However, in this study, none had type 1 diabetes and only one case had a history of type 2 diabetes; the relationship between diabetes and GO was difficult to be evaluated. Insulin resistance is the basic component of type 2 diabetes [24] thus, we counted patients with prediabetes and diabetes as a group and assessed the impact of insulin resistance on GO instead. Our data showed that (1) diabetes plus prediabetes, fasting plasma glucose, and HOMA-IR were associated with the presence of GO; and (2) fasting plasma glucose, fasting plasma insulin, and HOMA-IR levels were positively correlated with the severity of GO. After correcting for the age, gender, titers of TSHR antibodies, and steroid use, we found that fasting plasma insulin and HOMA-IR remained associated with GO. These findings highlighted the importance of insulin resistance in the occurrence of GO. Supporting this notion, previous research had shown that Enalapril, an angiotensin-converting enzyme inhibitor with the ability to reduce insulin resistance [44,45], can also decrease the proliferation and production of hyaluronic acid in orbital fibroblasts from patients with GO [46]. These findings support our results. The underlying mechanisms linking obesity, insulin resistance, and GO had not been fully understood. Shared chronic inflammation and increased oxidative stress states among these diseases might explain the link [10,21]. Obesity is generally thought to be a state of chronic low-grade inflammation, which eventually leads to oxidative stress and insulin resistance [12,47]. BMI is associated with the level of hs-CRP [13]. A higher grade of inflammation had been found to be associated with a higher degree of insulin resistance and oxidative stress in patients with rheumatoid arthritis [17]. Oxidative stress is a possible cause of insulin resistance and GO, both of which can be improved by antioxidants [48,49,50,51,52,53]. Compatible with the aforementioned studies, we also found that hs-CRP levels were positively correlated with the severity of GO (rho = 0.299, *p* = 0.006). Our results supported chronic inflammation theory [54], in that hs-CRP levels [54] were positively correlated with the severity of GO. Compatible with this notion, a large nationwide database study reported a 40% reduction in hazard ratios in patients who received statin therapy [55], which may be because statins reduce plasma levels of C-reactive protein [56]. 

TSHR and insulin-like growth factor 1 (IGF-1 R) play essential roles in GO development [1,9,10]. Obviously, IGF-1 R is more important than TSHR in the treatment of GO because the inhibition of IGF-1 R with IGF-1 R antibody (Teprolumumab) had the best therapeutic effect ever seen [57,58]. Insulin can bind to IGF-1 R and activate biological effects [59]. It can explain that hyperinsulinemia is associated with the presence of GO and correlated with the severity of GO in our analysis. This association may co-exist or be the relation of cause and effect. If insulin resistance is the important cause of GO in those obese patients, drugs to reduce insulin resistance, such as metformin [60], may be a possible treatment for GO.

The limitations of this study include the small number of cases without randomization and not being well designed which induces bias, cross-sectional study design without a follow-up. Our study did not do a computed tomography scan [1], which may have an impact on obesity-related factors and insulin resistance in GO.

## 5. Conclusions

In the present study, obesity-related factors were associated with the presence of GO and were correlated with the severity of GO. These relationships may be induced through the effects of high insulin levels possibly through IGF-1 R, insulin resistance, or hs-CRP. Although a larger prospective study is warranted to confirm the relationship, surveillance of the development of GO in patients with GD should be considered, particularly for those with obesity-related factors, such as higher fasting plasma insulin and HOMA-IR.

## Figures and Tables

**Table 1 medicina-58-01748-t001:** Clinical characteristics and laboratory data stratified by the presence or absence of Graves’ orbitopathy.

	Total	Absence of GO	Presence of GO	*p*
*N*	84	42	42	*---*
Age (year)	42 ± 12	41.6 ± 12.2	42.6 ± 11.5	0.700
Male/female (*n*)	20/64	10/32	10/32	1
Antithyroid drug treatment period of Graves’ disease (year)	6.1 ± 6.7	6.6 ± 5.8	5.6 ± 7.5	0.200
BMI (kg/m^2^)	22.7 ± 3.2	22.0 ± 2.7	23.4 ± 3.4	0.034
Central obesity (−/+)	63/21	32/10	31/11	0.801
Smoking (−/+)	65/19	36/6	29/13	0.068
Co-morbidity (−/+)				
Diabetes + prediabetes	70/14	39/3	31/11	0.019
Hypertension	78/6	41/1	37/5	0.202
Hyperlipidemia	78/6	42/0	36/6	0.026
Treatment (−/+)				
Anti-thyroid drugs	1/83	0/42	1/41	1
Thyroidectomy	76/8	37/5	39/3	0.713
Iodine-131	82/2	40/2	42/0	0.494
Steroid use	81/3	42/0	39/3	0.241
Thyroid dysfunction (−/+)	64/20	33/9	31/11	0.608
Titers of TSHR antibodies (%)	38.3 ± 26.4	33.8 ± 25.2	42.9 ± 27.3	0.112
Fasting plasma glucose (mg/dL)	92 ± 9	88.9 ± 8.5	94.6 ± 9.8	0.005
Fasting plasma insulin (uIU/mL)	5.9 ± 5.7	4.7 ± 2.9	7.1 ± 7.5	0.081
HOMA-IR	1.4 ± 1.3	1.1 ± 0.7	1.7 ± 1.7	0.045
hs-CRP (mg/dL)	0.1 ± 0.2	0.1 ± 0.2	0.2 ± 0.3	0.007
Severity of GO (−/+) ^§^				
Absent	42/42	0/42	42/0	
Mild	55/29	42/0	13/29	
Moderate to severe	71/13	42/0	29/13	

Data are presented as mean (±SD) or number. The differences between groups were evaluated using the two-sample t-test, Mann–Whitney-U test, chi-square test, or Fisher’s exact test. ^§^ According to EUGOGO classification, GO, Graves’ orbitopathy; BMI, body mass index; TSHR, thyroid-stimulating hormone receptor; HOMA-IR, homeostasis model assessment-estimated insulin resistance; hs-CRP, highly sensitive C-reactive protein.

**Table 2 medicina-58-01748-t002:** Associations between obesity-related factors and Graves’ orbitopathy among patients with Graves’ disease (*N* = 84) in a multivariate logistic regression analysis.

	Model 1		Model 2		Model 3		Model 4	
	Odds Ratio (95% CI)	*p*	Odds Ratio (95% CI)	*p*	Odds Ratio (95% CI)	*p*	Odds Ratio (95% CI)	*p*
BMI	1.172 (1.001–1.372)	0.048 *	1.173 (0.997-1.379)	0.054	1.186 (1.007–1.398)	0.042 *	1.182 (1.003–1.393)	0.046 *
Diabetes + prediabetes	3.011 (0.840–10.979)	0.091	2.936 (0.807–10.681	0.102	2.416 (0.643–9.076)	0.192	1.890 (0.470–7.591)	0.370
Hyperlipidemia	1.886 × 10^9^ (0.000-)	0.999	2.733 × 10^9^ (0.000-)	0.999	3.839 × 10^9^ (0.000-)	0.999	3.672 × 10^9^ (0.000-)	0.999
Fasting plasma glucose	1.071 (1.012–1.132)	0.017 *	1.07 (1.011–1.132)	0.019 *	1.060 (1.001–1.123)	0.046 *	1.057 (0.995–1.123)	0.070
Fasting plasma insulin	1.161 (1.010–1.335)	0.035 *	1.162 (1.005–1.344)	0.043 *	1.178 (1.012–1.372)	0.035 *	1.165 (1.001–1.355)	0.048 *
HOMA-IR	2.005 (1.113–3.611)	0.021 *	2.002 (1.084–3.696)	0.027 *	2.081 (1.105–3.921)	0.023 *	1.985 (1.046–3.764)	0.036 *
hs-CRP	8.476 (0.582–123.500)	0.118	8.700 (0.518–146.042)	0.133	8.875 (0.533–147.813)	0.128	7.950 (0.346–182.869)	0.195

Model 1: corrected by age and sex. Model 2: corrected by age, sex, and smoking. Model 3: corrected by age, sex, smoking, and the titer of TSHR antibodies. Model 4: corrected by age, sex, smoking, the titer of TSHR antibodies, and steroid usage. CI, confidence interval; BMI, body mass index; HOMA-IR, homeostasis model assessment-estimated insulin resistance; hs-CRP, highly sensitive C-reactive protein, OR, odds ratio, TSHR, thyroid-stimulating hormone receptor. * *p* < 0.05.

**Table 3 medicina-58-01748-t003:** Correlations between obesity-related factors and severity of Graves’ orbitopathy among patients with Graves’ disease (*N* = 84).

	Spearman’s Correlation Coefficients	*p*
BMI	0.285	0.009
Fasting plasma glucose	0.298	0.006
Fasting plasma insulin	0.243	0.026
HOMA-IR	0.270	0.013
hs-CRP	0.299	0.006

BMI, body mass index; HOMA-IR, homeostasis model assessment-estimated insulin resistance; hs-CRP, highly sensitive C-reactive protein.

## Data Availability

The Appendix A used to support the findings of this study are included within the Appendix A.

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
