# Peer review of "The Relationship between Obesity-Related Factors and Graves’ Orbitopathy: A Pilot Study"

_medicina, 2022, doi:10.3390/medicina58121748_

Round 1

Reviewer 1 Report (Previous Reviewer 1)

The manuscript had now been revised carefully. All the queries of the reviewer had been considered

Reviewer 2 Report (New Reviewer)

It is appropriate to publish the work.
For readers, it would be appropriate to give 95% confidence intervals for the Exp(B) statistics in Table 2.

This manuscript is a resubmission of an earlier submission. The following is a list of the peer review reports and author responses from that submission.

Round 1

Reviewer 1 Report

This is a very important study and the results may be hghly relavant for the treatment of patients with GO or better called thyroid associated ophthalmopathy (TAO). 

Queries:

1. The manuscript is well prepared, there are only only one point that has to be clarified page 3 line 12 ..but mostly half yeras (6.1 + 6.7 year) which also is writetn in Table 1. What does thsi mean?  Please clarify. 

2. Unfortunately the authors have been oberlooking one of the most exciting detection: Obviously not TSH bit IGF-1 plays the main role in TAO development, and the inhibition of IGF-1 with Teprolumumab and IGF-1 R antibody had the best therapeutic effect ever seen (Smith JT J Neuroophtalmol 2022 40:74-83 and Smith TJ eta al NEngl J Med 2017 376: 1748-61. These new insights perfectly explain the results of this study,  better, than those which are mentioned. Therefore the discussion has to be focussed on these new insights and a possible treatment could be also metformin in those obese patients.   

Reviewer 2 Report

Thank you for inviting me to review this paper about the association of obesity with Graves orbitopathy. There are some improvements that should be made prior to publishing.

1) Introduction - the final statement on lines 58-61 should include a more comprehensive sentence about the purpose of the study.

2) Methods - would be helpful to include an inclusion and exclusion criteria as well as more information about how patients were collected in order to minimize bias. Was it retrospective? 

3) Discussion - lines 176-177 - needs clarification. the rationale for how smoking was a risk factor does not entirely make sense.

4) Lines 196 - 198 could be rewritten to be clear for the reader

5) The limitations section is a bit weak and could be strengthen to cover the bias that was present in the study